# Relationship between Modern ART Regimens and Immunosenescence Markers in Patients with Chronic HIV Infection

**DOI:** 10.3390/v16081205

**Published:** 2024-07-26

**Authors:** Rusina Grozdeva, Daniel Ivanov, Dimitar Strashimirov, Nikol Kapincheva, Ralitsa Yordanova, Snejina Mihailova, Atanaska Georgieva, Ivailo Alexiev, Lyubomira Grigorova, Alexandra Partsuneva, Reneta Dimitrova, Anna Gancheva, Asya Kostadinova, Emilia Naseva, Nina Yancheva

**Affiliations:** 1Department of Infectious Diseases, Parasitology and Tropical Medicine, Medical University Sofia, 1431 Sofia, Bulgaria; dannieltiv@gmail.com (D.I.); dstrashimirov@yahoo.com (D.S.); nicole.kyuchukova@gmail.com (N.K.); yordanova_r@abv.bg (R.Y.); 2Central Laboratory of Clinical Immunology, University Hospital Alexandrovska, 1431 Sofia, Bulgaria; sneji_jm@yahoo.com (S.M.); neiseria2@gmail.com (A.G.); 3National Reference Laboratory of HIV, National Center of Infectious and Parasitic Diseases (NCIPD), 1504 Sofia, Bulgaria; ivoalexiev@yahoo.com (I.A.); lyubomiragrigorova@gmail.com (L.G.); alexandra.partsuneva@gmail.com (A.P.); naydenova.reneta@gmail.com (R.D.); gancheva.anna@gmail.com (A.G.); deshova.asi@gmail.com (A.K.); 4Department of Health Economics, Faculty of Public Health “Prof. Tsekomir Vodenicharov, MD, DSc”, Medical University of Sofia, 1527 Sofia, Bulgaria; e.naseva@foz.mu-sofia.bg; 5Medical Faculty, Sofia University St. Kliment Ohridski, 1407 Sofia, Bulgaria

**Keywords:** aging, HIV-1, combined antiretroviral therapy, chronic inflammation

## Abstract

The increased life expectancy of PLHIV (People Living with HIV) and the successful highly combined antiretroviral therapy (cART) poses new clinical challenges regarding aging and its co-morbid condition. It is commonly believed that HIV infection “accelerates” aging. Human immunodeficiency virus type 1 (HIV-1) infection is characterized by inflammation and immune activation that persists despite cART, and that may contribute to the development of co-morbid conditions. In this regard, we aimed to compare current cART regimens in light of premature aging to evaluate differences in their ability to reduce immune activation and inflammation in virologically suppressed patients. We studied a panel of biomarkers (IFN-γ, IL-1β, IL-12p70, IL-2, IL-4, IL-5, IL-6, IL-13, IL-18, GM-CSF, TNF-α, C-reactive protein, D-dimer, soluble CD14), which could provide a non-invasive and affordable approach to monitor HIV-related chronic inflammation. The results of the current study do not provide hard evidence favoring a particular cART regimen, although they show a less favorable regimen profile containing a protease inhibitor. Our data suggest an incomplete reduction of inflammation and immune activation in terms of the effective cART. It is likely that the interest in various biomarkers related to immune activation and inflammation as predictors of clinical outcomes among PLHIV will increase in the future.

## 1. Introduction

In 2022, almost three-quarters (71%) of people living with HIV (PLHIV) have undetectable viral load thanks to cART. Viral suppression enables PLHIV to live long and healthy lives [1]. The proportion of HIV-1 patients over the age of 50 has increased dramatically due to improved survival and in relation to the higher number of infected people in older age groups [2]. We must not forget that these patients still have increased morbidity and mortality compared to the general population [3], even upon completion of successful cART. PLHIV show progressive immune system dysfunction similar to that observed in the elderly; that is why HIV infection is considered to be a model of accelerated immunosenescence [4].

All that raises new clinical issues regarding aging and its related co-morbid conditions in PLHIV [5], which occur more frequently in them than in the general population [6,7] and in relatively young HIV-1 patients even with persistent viral suppression. They include neurocognitive disorders, cardiovascular disease (CV), metabolic syndrome, bone abnormalities, and non-HIV cancers [8]. Most of these diseases are associated with chronic inflammation and activation of the immune system [9].

A broad range of biomarkers has been proposed for the study of aging. For example, telomere length, epigenetic clocks, mitochondrial DNA, apolipoprotein J/Clusterin, proteasome subunits, NAD/NADH ratio, lipid alterations in plasma, plasma levels of essential amino acids, soluble inflammatory markers, cell surface molecules, and MARK-AGE. Future studies would focus on the optimization of biomarkers for use in routine clinical practice [10].

Microbial translocation, continued production of HIV, co-infections, loss of regulatory T cells, damage to the thymus and lymphoid infrastructure, smoking, obesity, alcohol drinking, and low social status are indicated as causes of aging [11,12,13,14]. The progressive mitochondrial damage induced by cART treatment likely contributes to senescence even in the era of modern cART [15].

This made us set an objective to compare different cART regimens in PLHIV in the light of residual immune activation and dysfunction by examining a panel of biomarkers (IFN-γ, IL-1β, IL-12p70, IL-2, IL-4, IL-5, IL-6, IL-13, IL-18, GM-CSF, TNF-α, C-reactive protein, D-dimer, soluble CD14—a part of proposed biomarkers for immune activation), which could provide a non-invasive and affordable approach to monitor HIV-related chronic inflammation in virologically suppressed patients and help us determine whether there is a difference in the ability of cART regimens to correct immune activation in chronic HIV infection. This will probably reduce morbidity and mortality resulting from non-HIV-1 related diseases and the burden on the health system.

## 2. Materials and Methods

### 2.1. Participants’ Selection

It concerns a prospective cross-sectional study conducted in the Acquired Immunodeficiency Department at Infectious Disease Hospital “Prof. I. Kirov’’ EAD—Sofia, Bulgaria.

We involved a total of 80 individuals in the study: 17 of them taking Emtricitabine/Tenofovir alafenamide/Darunavir/Cobicistat (FTC/TAF/DRV/c); 17 of them taking Lamivudine/Tenofovir disoproxil fumarate/Doravirine (3TC/TDF/DOR); 18 of them taking Emtricitabine/Tenofovir alafenamide/Bictegravir (FTC/TAF/BIC); 18 of them taking Lamivudine/Dolutegravir (3TC/DTG); and 10 of them taking Dolutegravir/Rilpivirine (DTG/RPV). A description of the studied group is presented in Table 1 and Table 2.

The following inclusion criteria were used: positive HIV status; age over 18 years; all patients involved in the study are HIV-positive with persistent viral suppression (VL < 40 c/mL > 1 year).

The following exclusion criteria were used: age under 18 years; pregnancy; participants who refuse to sign informed consent; administration of non-steroidal anti-inflammatory drugs in the last 1 month; patients who do not have persistent viral suppression, i.e., who do not have VL < 40 c/mL > 1 year.

Each patient signs an informed consent approved by the research ethics committee at Sofia Medical University (ВК-455/31.03.2023).

### 2.2. Collection of Biological Material

For the purposes of biomedical research, the following serological markers, such as IFN-γ, IL-1β, IL-12p70, IL-2, IL-4, IL-5, IL-6, IL-13, IL-18, GM-CSF, TNF-α, C reactive protein, D-dimer, and CD14 soluble in venous blood, will be analyzed. A closed vacutainer system was used for each participant in the study, following the standard sterility procedures. We used vacutainers with a clot activator and 3–10 mL blood. To determine the viral load (VL), we used EDTA tubes as an anticoagulant.

Storage of biological material to study IFN-γ, IL-1β, IL-12p70, IL-2, IL-4, IL-5, IL-6, IL-13, IL-18, GM-CSF, TNF-α, and soluble CD14: after collection and centrifugation, equal aliquots of serum were frozen at a temperature of −80 °C until all samples were collected for their overall analysis. C reactive protein, D-dimer, and VL were examined simultaneously.

### 2.3. Biomarkers Measurement

For the purposes of the study, we used a highly sensitive immunological Human ProcartaPlex™ Kit (Termo Fisher, Vienna, Austria) to measure IFN-γ, IL-1β, IL-12p70, IL-2, IL-4, IL-5, IL-6, IL-13, IL-18, GM-CSF, and TNF-α proteins in combination, in the form of a multiplex panel that uses Luminex xMAP technology for protein detection and quantification. The ProcartaPlex immunological tests that we used were based on the sandwich ELISA principles. The capture antibody in the ProcartaPlex assay is conjugated to magnetic beads and is not adsorbed to the microplate well so that the ProcartaPlex assay reagents are floating freely in the solution. Similarly, soluble CD14 was assayed with the help of CD14 Human ProcartaPlex™ Simplex Kit (Thermo fisher), using the technology described above.

The LLOQ (Limit of Quantitation) is listed for each analyte in the kit description. Due to the fact that the LLOQ in pg/microl indicated by the manufacturer is “Determined in cell culture medium”, but we are working with serum, where much lower concentrations of the given analyte are expected, we have used the following analysis: Curve Fitting Analysis Point-to-Point- interpolation between two adjacent points using a line (y = ax + b). The parameter of the *x*-axis is concentration (in pg/mL), and the parameter of the y-axis is given as MFI. The MFI of each standard point is blankcorrected by subtraction of the Blank-MFI (MFI-MFI_Blank). This type of analysis allowed us to detect a value between the concentration of the lowest standard and corrected Blank-MFI. The calculation of LLOQ and ULOQ was based on the maximum acceptable Bias with a Cut-off at 30% Bias and Fit 100%. So, practically, we were able to detect values between 0 and the lowest standard value (specified by the manufacturer).

We used the Finecare™ CRP Rapid Quantitative Test and Finecare™ D-Dimer Rapid Quantitative Test, along with the Finecare™ FIA Meter—fluorescent immunoassay for quantitative measurement of D-Dimer. We used whole human blood as a material. It is based on fluorescence immunoassay technology. The Finecare™ D-Dimer Rapid Quantitative Test uses a sandwich immunodetection method. When the sample is added to the sample well, the fluorescently labeled anti-D-Dimer/anti-CRP detector on the membrane binds to the D-Dimer/CRP antigen in the blood. The mixture migrates onto the nitrocellulose matrix of the test strip, and the detector antibody and D-Dimer/CRP complexes are captured by the D-Dimer/CRP antibody that has been immobilized on the test strip. Thus, if a larger amount of D-Dimer/CRP antigen is contained in the sample, more complexes will accumulate on the test strip. The fluorescence intensity of the detector antibody reflects the amount of captured D-dimer/CRP, and the Finecare™ FIA Meter indicates the concentration of D-dimer/CRP. The result is displayed as XXX mg/L by the Finecare™ FIA Meter.

### 2.4. Viral Load Measurement

We used the Abbott RealTime HIV-1 test. The Abbott RealTime HIV-1 Test is an in vitro reverse transcription polymerase chain reaction (RT-PCR) test for quantification of human immunodeficiency virus type 1 (HIV-1) on the m2000 automated system in human plasma taken from HIV-1 infected individuals above 40 copies/mL Reverse transcription PCR is a qualitative test for determination of HIV-1 RNA in plasma. Real-time PCR uses real-time homogeneous fluorescence detection technology.

The design of the probes included in the test is partially double-stranded and allows the detection of a variety of group M subtypes and group O isolates. The assay was standardized based on a viral standard from the Virology Quality Assurance (VQA) system of the Laboratory of AIDS Clinical Trial Group [(Yen-Lieberman, 1996)] and in accordance with the first international HIV-1-RNA (97-656) standard of the World Health Organization (WHO) [(Holmes, 2001)]. HIV-1 viral load testing uses RT-PCR to generate an amplified product of the HIV-1 RNA genome in clinical samples. A sequence unrelated to the HIV-1 target sequence was introduced into each tube at the beginning of sample preparation. This unrelated RNA sequence was amplified along with the viral genome through RT-PCR and served as an internal control (IC) to demonstrate that the process was properly conducted in each sample. The amount of HIV-1 was measured at each cycle using fluorescently labeled oligonucleotide probes. Probes do not produce a signal unless they are associated with the target product.

The entire work process was performed according to the manufacturer’s recommendations.

### 2.5. Statistical Methods

The quantitative variables are presented with median and interquartile ranges (25th and 75th percentiles) due to their non-Gaussian distribution. The distribution pattern was evaluated using the Kolmogorov–Smirnov test. Spearman’s rank correlation coefficient was used to evaluate relationships between quantitative variables.

The mean values of two groups were compared with those obtained upon completion of the Mann–Whitney U test and mean values of more than two groups were compared with those obtained upon completion of the Kruskal–Wallis test; *p*-values < 0.05 were considered to be statistically significant. All analyses were performed using IBM SPSS Statistics 26.

## 3. Results

### 3.1. Relationship between the Studied Indicators and Some Variables

IL-12p70 was weakly positively related to the age (Spearman’s rho = 0.237, *p* = 0.034). A significant difference was found between the two sexes only for IL-4 (*p* = 0.002), which was significantly higher in men.The relationship between examined biomarkers and the baseline CD4 count and the clinical category of HIV was evaluated with Spearman’s rank correlation coefficient. No significantly different values of the indicators were proven depending on the baseline CD4 count. The baseline clinical category of HIV was weakly positively related to IL-18 (Sperman’s rho = 0.226, *p* = 0.044) and IL-4 (Sperman’s rho = 0.244, *p* = 0.029).No significantly different values of the indicators were proven depending on the presence of cardiovascular diseases (CVDs), sexually transmitted diseases (STDs), or other diseases.No significantly different values of the indicators were proven depending on the presence of smoking, although the differences in the values of sCD14 were higher in smokers but with borderline significance (*p* = 0.058). No significantly different values of the indicators were proven depending on the alcohol use.IFN-γ was found to be related to the duration of cART through a weak and inverse relationship, i.e., the increased duration of the therapy leads to a decreased value of IFN-γ (Spearman’s rho = −0.271, *p* = 0.015). IL-4 was moderately strongly positively related to the duration of cART (Spearman’s rho = 0.306, *p* = 0.006). D-dimer was weekly positively related to the duration of cART (Spearman’s rho = 0.228, *p* = 0.043).Significantly different values of the indicators were proven depending on whether the patient switched therapy in the values of IL-2 (*p* = 0.005) and IL-5 (*p* = 0.005)—patients who switched had higher values.We found some dependencies between the studied indicators and the duration of taking some of the cART regimens. IFN-γ was negatively related to the duration of the previous intake of integrase inhibitors (Sperman’s rho = −0.485, *p* = 0.049) and nucleoside analogs (Sperman’s rho = −0.404, *p* = 0.010). IL-2 was positively related to the duration of current non-nucleoside analogs use (Sperman’s rho = 0.442, *p* = 0.019). IL-2 was positively related to the total duration of nucleoside analogs use (Sperman’s rho = 0.259, *p* = 0.020) (Table 3).

### 3.2. Distribution of Biomarkers Depending on the cART Regimen

Lower measured values (almost null) of IFN gamma (*p* < 0.001) and IL12p70 (*p* = 0.006) were observed in DTG/RPV and FTC/TAF/DRV/c regimes compared to 3TC/DTG, 3RC/TDF/DOR, and FTC/TAF/BIC regimens (Figure 1 and Figure 2).With regards to IL12p70, only in the FTC/TAF/BIC regimen is the median distinctly higher, i.e., the patients from this group tend to show significantly higher values of this indicator compared to all other groups. Regarding the 3TC/DTG and 3TC/TFF/DOR regimens, the results are very similar (Figure 2).IL-18 (*p* = 0.009) results showed that in patients following NRTIs+NNRTI (3TC/TDF/DOR) and NRTIs+INI or NRTI+INI (FTC/TAF/BIC and 3TC/DTG) regimens, higher values than in those following INI+NNRTI (DTG/RPV) and NRTIs+PI (FTC/TAF/DRV/c) regimens were observed, although we reported increased values for the other two regimens as well (Figure 3).In FTC/TAF/BIC patients, the lowest variation in the IL-18 values was observed, and in patients in the given group, close values of the studied parameter were observed, with the range corresponding approximately to the mean values of the measured parameter. Q3 (75th percentile) for the 3TC/TDF/DOR regimen is higher than that reported in all other groups, i.e., 25% of the highest measured values of 3TC/TDF/DOR were higher than that reported in the other groups (Figure 3).Patients taking NRTIs+INI or NRTI+INI (FTC/TAF/BIC and 3TC/DTG) and NRTIs+NNRTI (3TC/TDF/DOR) showed the lowest IL-5 (*p* = 0.019) values (Figure 4).There was a trend toward measurement of increased IL-5 values in the DTG/RPV and FTC/TAF/DRV/c groups, with these values being slightly higher for the DTG/RPV regimen (Figure 4).

The highest sCD14 (*p* = 0.082) values were reported in the NRTIs+PI (FTC/TAF/DRV/c) regimen, i.e., in a regimen containing a protease inhibitor. We found the lowest values in patients following NRTIs+INI or NRTI+INI (FTC/TAF/BIC and 3TC/DTG) and NRTIs+NNRTI (3TC/TDF/DOR) regimens (Figure 5).

In the FTC/TAF/BIC regimen, the IQR is the widest, and the median is high, which shows markedly higher values of TNF-α (*p* = 0.007) compared to all other groups (Figure 6).

### 3.3. Distribution of Biomarkers According to Drug Class

We found significantly lower levels of IFN-γ (*p* < 0.001) in the protease inhibitor (darunavir-containing regimen) patients in the current therapy, lower levels of IL-18 (*p* = 0.023), and higher levels of sCD14 (*p* = 0.08) (Table 3).NNRTIs in current therapy are associated with lower levels of IL-1β (*p* = 0.048).

### 3.4. Relationships between the Studied Indicators

IL-6 was moderately strongly positively related to GM-CSF (Sperman’s rho = 0.329, *p* = 0.003), IFN-γ (Sperman’s rho = 0.387, *p* = 0.000), and IL-12p70 (Sperman’s rho = 0.35, *p* = 0.004).No relationship between CRP, d-dimer, and sCD14 with any of the series (*p* > 0.05) was proven.

## 4. Discussion

A lot of evidence on the presence of residual immune activation and inflammation in PLHIV was accumulated despite the successful cART and the associated risks of co-morbidity conditions and death. Evidence that despite cART, the levels of a number of immune inflammation markers remain increased was also accumulated [16].

Statistically significant differences were observed in the levels of pro- and anti-inflammatory circulating cytokines that stimulate and suppress the immune system, respectively, between HIV-infected and non-HIV-infected individuals [17]. The longitudinal assessment of circulating cytokines in one study showed a worse cytokine profile for cART-naïve patients compared to cART-treated patients and healthy controls. The data indicate that cART induces normalization of IL-4, IL-6, and IL-10. INF-γ, TGF-β, and TNF-α did not fully respond to cART after 1 year of therapy [16]. The Biomarker Cohort study showed a relationship between IL6, D-dimer, and soluble 14 (sCD14) and mortality in both HIV-positive and non-HIV-positive patients [18]. Data from other studies found that interleukin-6 [IL-6], C-reactive protein [hsCRP], soluble CD14 [sCD14], and D-dimer correlated with the overall mortality due to HIV-1 infection [19,20].

Studies comparing the modern cART regimens in light of the residual immune activation and dysregulation are of great interest [21]. For example, in one prospective study, cART-naive patients started treatment with RAL, ATV/r, or DRV/r with TDF/FTC. Patients who achieved viral suppression did not show a consistent pattern in terms of the studied markers (hsCRP, D-dimer, sCD14, and IL-6) favoring any of the cART regimens [22].

Data from some sources show a reduction in the microbial translocation markers in people randomized to raltegravir (RAL) compared to people randomized to non-nucleoside reverse transcriptase inhibitors (NNRTIs) [23,24,25]. Data from other sources show that the shift toward an intensification with RAL does not definitively demonstrate significant changes in systemic inflammation and immune activation [26,27].

Of the 13 markers examined, GM-CSF, IL-1β, IL-12p70, IL-2, IL-4, IL-5, IL-6, IL-13, IL-18, D-dimer, CRP, TNF-alpha, and sCD14 at INF-γ, IL12p70, IL-18, IL-5, TNF-alpha, and sCD14 we established a dependence with the cART regimen. The data show a progressive decrease, even not significant, in the baseline IFN-γ production with immunodeficiency progression [28]. cART significantly increased the plasma levels of IFN-γ [29,30] relative to the baseline levels. However, we found a reduction of IFN-γ when the duration of cART increased (Spearman’s rho = −0.271, *p* = 0.015), which we explained by the residual immune activation and dysregulation that had led to immunosenescence over time. The fact that regimens containing a protease inhibitor (DRV/c) or those without a nucleoside analog as the basis of the therapy (DTG/RPV) showed lower IFN-γ values in our study could define some cART regimens as more effective than others, but further studies with a larger number of participants need to be conducted. Of course, one biomarker is not sufficient for such an evaluation.

Exposure of monocyte-derived dendritic cells to plasma from untreated HIV-infected donors with chronic infection resulted in suppression of IL-12p70 secretion. Similar observations were made for TNFα. The suppressive effect was weaker in plasma donors following the cART regimen [31]. Our explanation is that cART regimens with better efficacy will have a weaker suppressive effect on IL-12 secretion. The regimens in our study are FTC/TAF/BIC, 3TC/DTG, and 3TC/TDF/DOR. According to other sources, IL-12p70 is increased in HIV-positive patients compared to healthy controls [32] and is more represented among patients taking a protease inhibitor compared to those taking an integrase inhibitor [33], which is inconsistent with our results.

Markedly low levels of IL-18 were observed in cART patients who achieved viral suppression [34], although our data did not establish such a dependence. The higher serum levels of this indicator may be a useful marker in HIV-1-infected patients with metabolic disorders and in fat redistribution, as well as a sensitive predictor of cardiovascular complications in treated patients [35]. None of the regimens in the present study showed complete normalization of IL-18 despite the observed success of cART. We report lower values of IL-18 for patients following the DTG/RPV regimen, although, with regards to the other markers, the relation for this regimen is inverse. This may be due to unreported co-morbidity conditions, for example.

Data show that virological non-responders have significantly increased IL-5 levels compared to cART responders. However, the values of this indicator do not completely normalize even in the case of durable viral suppression with high CD4 T lymphocyte levels [36]. According to our data, higher values of IL-5 are found in patients following FTC/TAF/DRV/c and DTG/RPV regimens.

On the one hand, IL-6 values are significantly increased in the case of HIV but show no significant difference compared to the control group participants after 1 year of therapy (*p* > 0.05), on the other hand [17]. One study shows higher IL-6 levels in patients taking a protease inhibitor compared to patients taking efavirenz and nevirapine, although this relation is not significant for atazanavir. IL-6 does not show significant differences depending on the cART regimen in our study. The higher levels of hsCRP and D-dimer are related to the higher levels of IL-6, according to some data [37]. According to our data, IL-6 is moderately strongly related to GM-CSF (Sperman’s rho = 0.329, *p* = 0.003), IFN-γ (Sperman’s rho = 0.387, *p* = 0.000), and IL-12p70 (Sperman’s rho = 0.315, *p* = 0.004). We do not prove any relationship between CRP, d-dimer, and sCD14 with any of the series (*p* > 0.05).

After the beginning of the cART therapy, IL-4 values gradually decrease to those comparable to the healthy controls [28]. We found that IL-4 was moderately strongly positively related to the duration of cART (Spearman’s rho = 0.306, *p* = 0.006), and the clinical category of HIV was weakly positively related to IL-4 (Sperman’s rho = 0.244, *p* = 0.029). In this regard, this marker may be potentially useful for assessing HIV progression.

Macrophage activation markers, such as sCD14, are associated with the risk of cardiovascular disease (CVD) in HIV-1-infected individuals [38,39,40]. We did not find such a relationship, but we did find it in smokers, but with borderline significance (*p* = 0.058). However, we did not find any relation between the studied markers and the presence of CVD. Data show that sCD14 decreases more markedly in patients following the RAL regimen compared to those following PI/r and NNRTI regimens [23,24]. These findings are consistent with our data.

The results of the current study do not provide hard evidence favoring a particular cART regimen, although they show a less favorable regimen profile containing a protease inhibitor. Moreover, we have to take into account some restrictions, such as the predominance of the male gender among the participants in the study. The design, and in particular, the cross-sectional study providing information on the tested group at a certain point in time, may also be considered to be a disadvantage, although some data show a stable level of various markers of systemic inflammation 1 year after starting HART therapy, which suggests that their re-evaluation may not be necessary [41]. The small number of participants and the lack of healthy controls, patients with VL > 40 c/mL but with a similar therapy history, and patients who do not take cART are also disadvantages due to our limited budget. All of the factors mentioned above may have influenced the results.

## 5. Conclusions

In general, our data suggest an incomplete reduction of inflammation and immune activation under effective cART conditions. These results also highlight the need for further study of different HART regimens to confirm whether a given regimen has a greater anti-inflammatory effect as a possible means of effective prevention of long-term co-morbid conditions in the cases of HIV-1 infection. It is likely that the interest in various biomarkers related to immune activation and inflammation as predictors of clinical outcomes among PLHIV will increase in the future.

## Figures and Tables

**Figure 1 viruses-16-01205-f001:**
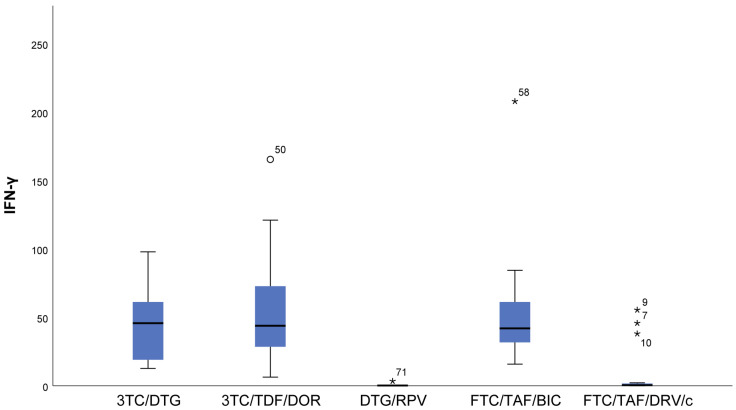
Distribution of IFN-γ depending on the cART regimen. Patients following DTG/RPV and FTC/TAF/DRV/c regimens had almost zero IFN-γ levels. IQR (interquartile range) is narrower for the FTC/TAF/BIC regimen than the 3TC/DTG and 3TC/TDF/DOR regimen, i.e., the values vary to the smallest degree. ***** The symbol * means extreme values. The symbol ° means outliers.

**Figure 2 viruses-16-01205-f002:**
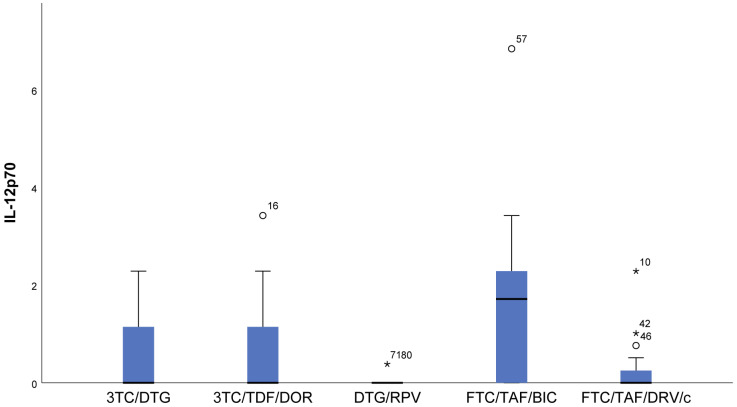
Distribution of IL12p70 depending on the cART regimen. The lowest IL-12p70 values were observed in DTG/RPV patients. They were followed by those following the FTC/TAF/DRV/c regimen. ***** The symbol * means extreme values. The symbol ° means outliers.

**Figure 3 viruses-16-01205-f003:**
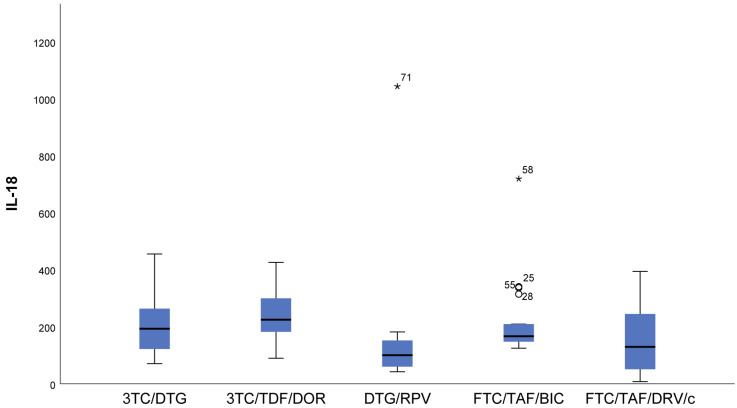
Distribution of IL-18 depending on the cART regimen. The FTC/TAF/BIC group falls within the narrowest IQR. In this group, we observed the smallest variation in IL-18 values. ***** The symbol * means extreme values. The symbol ° means outliers.

**Figure 4 viruses-16-01205-f004:**
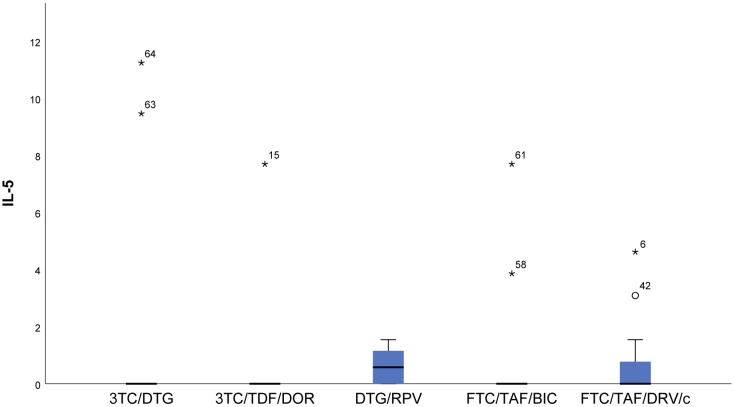
Distribution of IL-5 depending on the cART regimen. FTC/TAF/BIC, 3TC/DTG, and 3TC/TDF/DOR had the lowest IL-5 levels. One patient in the FTC/TAF/BIC group of the patients with significantly increased IL-5 values was found to be in advanced stages of immunodeficiency, although, at the time of the study, he had a satisfactory value of CD4 221 and type II DM. One patient in the FTC/TAF/BIC group was diagnosed with stage B of HIV infection and had very good values of CD4 854 at the time of the study and a co-morbid condition, duodenal ulcers. Both patients continued their treatment upon completion of another therapeutic regimen, and their ART regimen had a long duration—7.5 and 6 years, respectively. All other patients with significantly increased values were in stage A and had no known co-morbid conditions. ***** The symbol * means extreme values. The symbol ° means outliers.

**Figure 5 viruses-16-01205-f005:**
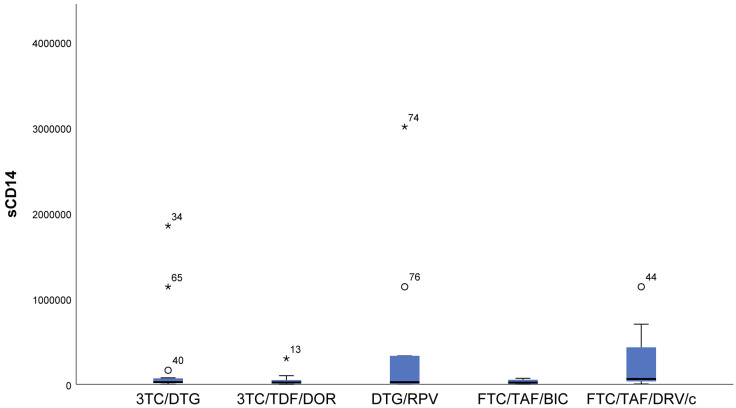
Distribution of sCD14 depending on the cART regimen. Increased values were observed in the regimen containing a protease inhibitor and in the two-drug regimen without nucleoside analog. ***** The symbol * means extreme values. The symbol ° means outliers.

**Figure 6 viruses-16-01205-f006:**
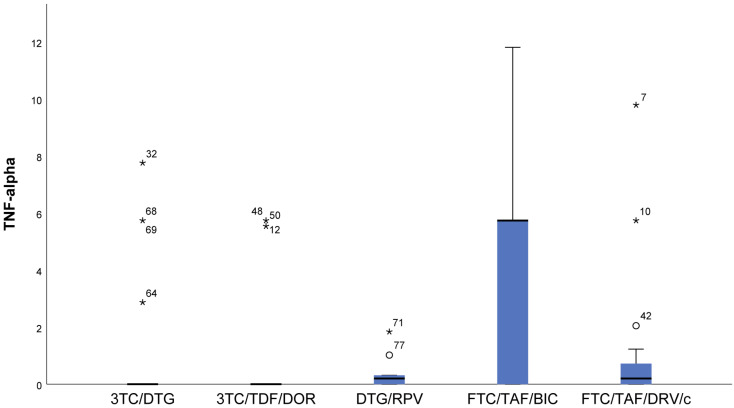
Distribution of TNF-α depending on the cART regimen. 3TC/DTG and 3TC/TDF/DOR regimens have the lowest values of TNF-α. ***** The symbol * means extreme values. The symbol ° means outliers.

**Table 1 viruses-16-01205-t001:** A description of the studied group.

	Mean	Standard Deviation	Median	Percentile 25	Percentile 75
Age	41	8			
Baseline CD4 count			370	252	537
Baseline CD8 count			1012	707	1290
Baseline CD4/CD8 count			0.37	0.23	0.55
Baseline VL copies/mL count			64,429	14,300	150,318
Baseline log10	4.69	0.92			
Duration of cART (years)			3.5	2.5	6.8
Present CD4	740	336			
Present CD8	836	297			
Present CD4/CD8			0.86	0.63	1.22
PI (currently)			3	3	3
INI (currently)			2.5	2.0	3.0
NNRTI (currently)			3	2	3
NRTI (currently)			2.5	2.0	3.0
PI (previous)			5	4	7
INI (previous)	3	1			
NNRTI (previous)			4	4	5
NRTI (previous)			4	3	5
PI (total)			4.5	3.0	7.8
INI (total)			3	3	4
NNRTI (total)			3	2	7
NRTI (total)			3.5	2.5	5.8

**Table 2 viruses-16-01205-t002:** A description of the studied group.

		Count	Column N %
Sex	Female	8	10.0%
	Male	72	90.0%
cART (Antiretroviral therapy)	3TC/DTG	18	22.5%
	3TC/TDF/DOR	17	21.3%
	DTG/RPV	10	12.5%
	FTC/TAF/BIC	18	22.5%
	FTC/TAF/DRV/c	17	21.3%
Switched	no	40	50.0%
	yes	40	50.0%
Previous cART	3TC/ZDV+LPV/r	1	2.5%
	ABC/3TC/DTG	8	20.0%
	ABC/3TC+ATV+RTV	1	2.5%
	ABC/3TC+DRV/c	2	5.0%
	ABC/3TC+DRV+RTV	2	5.0%
	ABC/3TC+RAL	1	2.5%
	ABC/3TC+RPV	2	5.0%
	DDI+ZDV+LPV/r	1	2.5%
	FTC/TDF+DRV/c	2	5.0%
	FTC/TDF+DRV+RTV	5	12.5%
	FTC/TDF+DTG	7	17.5%
	FTC/TDF+RAL	1	2.5%
	FTC/TDF+RPV	6	15.0%
	TDF+EFV	1	2.5%
HIV stages at cART initiation	CD4 over 500 cells/µL	31	38.8%
	CD4 from 250 to 499 cells/µL	33	41.3%
	CD4 under 250 cells/µL	16	20.0%
Duration of VL < 40	to 3 years	38	47.5%
	from 3 to 5 years	14	17.5%
	from 5 to 10 years	28	35.0%
Clinical presentation	A	48	60.0%
	B	18	22.5%
	C	14	17.5%
CVDs	no	72	90.0%
	yes	8	10.0%
STDs	no	53	66.3%
	yes	27	33.8%
Others	no	65	81.3%
	yes	15	18.8%
Smoking	no	48	60.0%
	yes	32	40.0%
Psychoactive drugs	no	76	95.0%
	yes	4	5.0%
Alcohol	no	76	95.0%
	yes	4	5.0%
Other harmful factors	no	79	98.8%
	yes	1	1.3%
Harmful factors	no	42	52.5%
	yes	38	47.5%
Transmission group	IVDU	9	11.3%
	MSM	58	72.5%
	unknown	13	16.3%
PIs in present therapy	no	63	78.8%
	yes	17	21.3%
INIs in present therapy	no	34	42.5%
	yes	46	57.5%
NNRTIs in present therapy	no	53	66.3%
	yes	27	33.8%
NRTIs in present therapy	no	10	12.5%
	yes	70	87.5%
PIs in previous therapy	no	27	67.5%
	yes	13	32.5%
INIs in previous therapy	no	23	57.5%
	yes	17	42.5%
NNRTIs in previous therapy	no	31	77.5%
	yes	9	22.5%
NRTIs in previous therapy	no	0	0.0%
	yes	40	100.0%

**Table 3 viruses-16-01205-t003:** Dependencies between the studied indicators and the duration of taking some of the cART regimens (PIs—Protease inhibitors, INIs—Integrase inhibitors, NNRTIs—Non-nucleoside reverse transcriptase inhibitors, NRTIs—Nucleoside reverse transcriptase inhibitors).

		GM-CSF	IFN-γ	IL-1β	IL-12p70	IL-13	IL-18	IL-2	IL-4	IL-5	IL-6	TNF-alpha	CRP	d-dimer	sCD14
Duration of rho cART (years)	pho	0.116	−0.271	0.030	−0.069	−0.087	−0.067	0.266	0.164	0.306	0.057	0.146	0.058	0.228	0.148
	p	0.305	0.015	0.793	0.545	0.442	0.556	0.017	0.147	0.006	0.615	0.197	0.610	0.043	0.219
	N	80	80	80	80	80	80	80	80	80	80	80	79	79	71
PI (сurrently)	pho	0.459	0.266	−0.379	−0.017	0.149	0.190	−0.415	−0.224	−0.308	0.119	0.246	0.014	0.199	0.205
	p	0.064	0.302	0.134	0.948	0.567	0.465	0.097	0.387	0.229	0.648	0.340	0.958	0.444	0.481
	N	17	17	17	17	17	17	17	17	17	17	17	17	17	14
INI (сurrently)	pho	−0.194	−0.184	−0.044	−0.171	−0.115	−0.047	−0.140	−0.157	−0.008	−0.326	0.076	−0.101	0.107	−0.046
	p	0.201	0.226	0.775	0.262	0.452	0.761	0.360	0.302	0.959	0.029	0.618	0.514	0.490	0.777
	N	45	45	45	45	45	45	45	45	45	45	45	44	44	41
NNRTI (сurrently)	pho	−0.201	−0.028	0.319	0.078	−0.113	0.069	0.442	0.247	0.099	0.239	0.438	0.046	0.254	0.004
	p	0.306	0.886	0.098	0.691	0.566	0.726	0.019	0.205	0.617	0.220	0.020	0.815	0.193	0.983
	N	28	28	28	28	28	28	28	28	28	28	28	28	28	26
NRTI (сurrently)	pho	−0.005	−0.097	−0.037	−0.101	−0.159	−0.048	−0.018	−0.053	−0.101	−0.081	0.109	−0.026	0.200	0.116
	p	0.967	0.425	0.758	0.405	0.189	0.694	0.880	0.664	0.405	0.504	0.371	0.830	0.099	0.373
	N	70	70	70	70	70	70	70	70	70	70	70	69	69	61
PI (previous)	pho		−0.207	−0.407	−0.212	0.020	0.342	−0.180	0.472	−0.083	−0.244	0.124	0.004	−0.131	−0.231
	p		0.477	0.149	0.467	0.945	0.232	0.539	0.089	0.778	0.400	0.673	0.988	0.656	0.427
	N	14	14	14	14	14	14	14	14	14	14	14	14	14	14
INI (previous)	pho	0.038	−0.485	0.015	−0.262	−0.165	−0.465	0.078	−0.174	0.132	−0.191	−0.250	−0.133	0.133	0.439
	p	0.885	0.049	0.954	0.310	0.528	0.060	0.766	0.505	0.614	0.463	0.333	0.610	0.612	0.117
	N	17	17	17	17	17	17	17	17	17	17	17	17	17	14
NNRTI (previous)	pho	0.210	−0.402	0.349	−0.112		−0.332	0.131	0.420	0.436	0.490	−0.152	−0.443	−0.004	−0.110
	p	0.588	0.284	0.357	0.774		0.383	0.738	0.261	0.241	0.181	0.696	0.233	0.991	0.795
	N	9	9	9	9	9	9	9	9	9	9	9	9	9	8
NRTI (previous)	pho	−0.076	−0.404	−0.030	−0.254	−0.043	−0.194	0.051	0.136	0.138	−0.107	−0.179	−0.106	0.137	0.078
	p	0.643	0.010	0.853	0.114	0.792	0.231	0.753	0.404	0.395	0.510	0.269	0.517	0.399	0.649
	N	40	40	40	40	40	40	40	40	40	40	40	40	40	36
PI (total)	pho	−0.015	−0.029	0.043	0.010	0.316	0.139	0.102	0.352	0.115	0.007	0.161	−0.010	0.312	0.055
	p	0.944	0.893	0.843	0.964	0.133	0.516	0.634	0.092	0.593	0.973	0.453	0.964	0.138	0.814
	N	24	24	24	24	24	24	24	24	24	24	24	24	24	21
INI (total)	pho	0.101	0.057	0.009	0.095	0.003	0.123	0.117	0.148	0.214	0.022	0.153	−0.072	0.077	0.179
	p	0.503	0.707	0.952	0.532	0.984	0.414	0.438	0.328	0.154	0.886	0.309	0.639	0.613	0.256
	N	46	46	46	46	46	46	46	46	46	46	46	45	45	42
NNRTI (total)	pho	0.255	−0.249	0.197	−0.050	−0.371	−0.176	0.326	0.056	0.320	0.158	0.265	0.111	0.194	−0.172
	p	0.200	0.211	0.325	0.804	0.057	0.380	0.097	0.783	0.103	0.431	0.181	0.581	0.333	0.411
	N	27	27	27	27	27	27	27	27	27	27	27	27	27	25
NRTI (total)	pho	0.123	−0.181	0.070	−0.011	−0.061	−0.003	0.259	0.195	0.270	0.099	0.153	0.055	0.252	0.160
	p	0.278	0.108	0.537	0.924	0.590	0.979	0.020	0.083	0.015	0.384	0.177	0.632	0.025	0.184
	N	80	80	80	80	80	80	80	80	80	80	80	79	79	71

## Data Availability

Data are contained within the article.

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
