# Peer review of "Relationship between Modern ART Regimens and Immunosenescence Markers in Patients with Chronic HIV Infection"

_viruses, 2024, doi:10.3390/v16081205_

Round 1

Reviewer 1 Report

Comments and Suggestions for Authors

This study aimed to reveal the impacts of different ART regimens on HIV-related chronic inflammation. However, the data are not convincing due to caveats in the experimental setup. The following major concerns must be addressed:

Point #1: As the authors stated, many factors could contribute to immune activation during HIV persistence with ART besides the possible impact of ART itself. These factors include microbial translocation, residual HIV, co-infections (e.g., CMV), loss of regulatory T cells, damage to the thymus and lymphoid infrastructure, and smoking. However, there is no description about how these variables were controlled, whether these criteria were matched between different ART regimen groups. The results showed some immune factors are higher related to particular regimen, the findings could be related to other variables but not the ART regimen.

Point #2: In Table 1, the data showed that the duration of ART may affect some inflammation markers like IFN-γ and IL-2. However, for the comparison between different regimen groups, there is no data showing the duration time for each participant in different groups. This inconsistency raises concerns about the data presented later (Figure 1-3), making the results unconvincing.

Point #3: The inclusion criteria showed that all 78 PHIV had persistent viral suppression (VL < 40 c/ml for over a year). However, what about the longitudinal effects of possible HIV replication before this one-year period? I believe that immune activation induced by HIV expression in some immune cells could last more than a year.

Minor point: The authors tried to state the chronic immune activation persists in PHIV, it would be good to have a group of a HIVneg controls

Author Response

Comments 1: Point #1: As the authors stated, many factors could contribute to immune activation during HIV persistence with ART besides the possible impact of ART itself. These factors include microbial translocation, residual HIV, co-infections (e.g., CMV), loss of regulatory T cells, damage to the thymus and lymphoid infrastructure, and smoking. However, there is no description about how these variables were controlled, whether these criteria were matched between different ART regimen groups. The results showed some immune factors are higher related to particular regimen, the findings could be related to other variables but not the ART regimen.

Response 1: Thank you for pointing this out. Information will be added regarding whether and how certain variables affected the values of the investigated markers so that there is no doubt about the results obtained. In addition to data on the relationship of indicators with the duration of ART, we also include data on the relationship of indicators with age, sex, clinical manifestation of HIV, sexually transmitted diseases, other accompanying diseases, and some harms. This change can be found – page 7, linе 175-197.  

Comments 2: Point #2: In Table 1, the data showed that the duration of ART may affect some inflammation markers like IFN-γ and IL-2. However, for the comparison between different regimen groups, there is no data showing the duration time for each participant in different groups. This inconsistency raises concerns about the data presented later (Figure 1-3), making the results unconvincing.

Response 2: We agree that data showing duration times for each participant in different groups need to be presented. This change can be found – page 7, paragraph 6, line 198-200; page 7, paragraph 7, page 201-207; page 8, table 3.

Comments 3:  Point #3: The inclusion criteria showed that all 78 PHIV had persistent viral suppression (VL < 40 c/ml for over a year). However, what about the longitudinal effects of possible HIV replication before this one-year period? I believe that immune activation induced by HIV expression in some immune cells could last more than a year.

Response 3:  As I understand it, you are suggesting that the results may have been affected by HIV replication before viral suppression was achieved. For this reason, the study included patients who were evenly distributed in the different ART regimens according to the duration of HIV, accompanying diseases, harms, stages of HIV at the time of the study, and when the HIV infection was detected. The definition of sustained viral suppression was adopted as an inclusion criterion, but the study cohort mostly had a much longer duration of VL < 40 c/ml. In fact, the median VL below 40 for the study group was 3.5 years. The median baseline CD4 count was 370. The studied cohort was mostly in stages A or B of HIV infection and, respectively, stages 1 or 2 according to the CD4 count at the diagnosis of HIV infection. With this, we have tried as much as possible to clear the possible lingering influence of the HIV virus. Data in the literature generally show an improvement in a number of inflammatory markers during the course of cART. However, it is quite possible that the immune activation induced by HIV expression lasts for more than a year and that there is residual immune activation for a very long period despite successful cART. If this were the cause of the results of the above study, we would not have found dependencies of some markers with certain cART regimens. Data in the literature indicate residual immune activation in cART patients, but the question that my colleagues and I are more interested in is whether this activation is different for different cART regimens. This is also the basis of the present study.

Minor point 1:  The authors tried to state the chronic immune activation persists in PHIV, it would be good to have a group of a HIVneg controls.

Response :  The present study addresses a current and still unresolved issue related to the topic of premature aging in the course of chronic HIV infection. Our results agree with some literature data and not with others. We do not draw firm conclusions in view of the available limitations, and readers are advised not to be misled. The results provide grounds for further and more in-depth research on the topic. We regret the lack of HIV-negative controls, which is dictated by our limited resources.

4. Response to Comments on the Quality of English Language

Comment: English language fine. No issues detected.

Reviewer 2 Report

Comments and Suggestions for Authors

Grozdeva et al. aim "to compare different cART regimens in PLHIV in light of residual immune activation and dysfunction by examining a panel of biomarkers."

Major concerns and comments:

  1. A profound characterization of the population under study is mandatory for a more appropriate interpretation of the results. How long have the patients been under therapy? What are their CD4+ T cell counts, irrespective of the viral load (VL)? Considering the study's objective, besides evaluating HIV-treated patients with viral suppression, it should also include - at least - HIV patients without such suppression but with a similar therapy history. This will provide more robust data for the Discussion, as the current form appears to be a mere comparison with other references.

  2. According to the WHO Global HIV Programme, "the toxicity issues and adverse events associated with antiretrovirals are currently identified only intermittently and are not widely reported." However, the toxicity of different treatment schemes should be evaluated sharply using tests currently available in clinical labs for liver and kidney functional evaluation, at a minimum.

  3. The "inflammaging" evaluation during chronic immune activation includes multiple cellular markers besides those studied by the authors (T cell: CD28, CD57, KLRG1, PD-1, CD45RA/RO; B cell: CD27, IgD, CD19; NK cells: CD16; monocytes: CD14/CD16; myeloid cells: CD11b; cellular senescence markers such as SA-β-Gal (Senescence-Associated Beta-Galactosidase), p16^INK4a, p21^CIP1/WAF1, γ-H2AX). Consequently, the analysis should be interpreted as partial, and this consideration should be discussed.

  4. The units of soluble factors measured are absent in the graphics. The LLOD (Lower Limit of Detection) would also be useful for interpretation.

  5. Statistical values obtained should be included in the text.

  6. As mentioned before, besides the authors' included limitations ("The small number of participants and the lack of healthy controls and patients who do not take HART"), it is necessary to study PLWH with matched-history of therapy but without viral suppression.

  7. Minor comments

a. Check the cART expressions (also appear as "cAPT", "CART").

b. Tables 3, 4, and 5 may be shorten, just including in the text the relevant values and its statistics.

Comments on the Quality of English Language

Only minor amendments are needed.

Author Response

Comments 1: A profound characterization of the population under study is mandatory for a more appropriate interpretation of the results. How long have the patients been under therapy? What are their CD4+ T cell counts, irrespective of the viral load (VL)? Considering the study's objective, besides evaluating HIV-treated patients with viral suppression, it should also include - at least - HIV patients without such suppression but with a similar therapy history. This will provide more robust data for the Discussion, as the current form appears to be a mere comparison with other references.

Response 1: Thanks for the remark related to the characteristic of the investigated pollation. We will do our best to provide more details.

This change can be found – page 2-4, Table 1 and Table 2 in the revised manuscript.

As mentioned in the text, to our great regret, our resources were limited. There are data in the literature comparing residual immune activation between PHIV who are VL < 40 c/ml and those with VL > 40 c/ml, with the data predominantly favoring virologically suppressed patients. Given this and in light of the current UNAIDS 95-95-95 goals, we decided it would be more appropriate and relevant to compare not so much patients with VL < 40 c/ml and those with VL > 40 c/ml (such comparisons are numerous and examples are given in the presented text), but rather to ask ourselves which of the currently used cART regimens affect residual immune activation to a greater extent in the presence of viral suppression. Over 95% of those registered in our center (just over 1,500) are taking cART and over 95% are virologically suppressed, which on the one hand would create difficulties in selecting a group with suboptimal viral suppression but with a similar history of therapy and, on the other hand, was unwarranted given the fact that the problem of viral suppression is largely resolved for our center.

Comments 2: According to the WHO Global HIV Programme, "the toxicity issues and adverse events associated with antiretrovirals are currently identified only intermittently and are not widely reported." However, the toxicity of different treatment schemes should be evaluated sharply using tests currently available in clinical labs for liver and kidney functional evaluation, at a minimum.

Response 2:  The current study did not assess cART toxicity or cART-related adverse effects but compared different modern cART regimens for their ability to affect HIV-related chronic inflammation. The introductory part of the presented text mentions the existing toxicity of cART. I apologize if this has caused any misunderstanding regarding the objectives of the survey. Due to the presence of ambiguity, the text will be corrected. This change can be found – page 1, paragraph 1, and linе 22 and page 2, paragraph 3, line 58-60 in the revised manuscript.

Comments 3:  The "inflammaging" evaluation during chronic immune activation includes multiple cellular markers besides those studied by the authors (T cell: CD28, CD57, KLRG1, PD-1, CD45RA/RO; B cell: CD27, IgD, CD19; NK cells: CD16; monocytes: CD14/CD16; myeloid cells: CD11b; cellular senescence markers such as SA-β-Gal (Senescence-Associated Beta-Galactosidase), p16^INK4a, p21^CIP1/WAF1, γ-H2AX). Consequently, the analysis should be interpreted as partial, and this consideration should be discussed.

Response 3: Thank you for pointing this out. Unfortunately, covering all possible markers of chronic inflammation is not economically feasible for our research center. It will be further explained in the text that the study covers only a fraction of the existing markers for evaliating inflammation during chronic immune activation.. This change can be found – page 2, paragraph 2, and linе 50-55 and page 2, paragraph 4, line 64 in the revised manuscript.

Comments 4:  The units of soluble factors measured are absent in the graphics. The LLOD (Lower Limit of Detection) would also be useful for interpretation.

Response 4:  The LLOQ (Limit of Quantitation)  is listed for each analyte in the kit description. Due to the fact that the LLOQ in pg/microl indicated by the manufacturer are "Determined in cell culture medi-um", but we are working with serum, where much lower concentrations of the given ana-lyte are expected, we have used the following analysis:

Curve Fitting Analysis Point-to-Point- interpolatation between two adjacent points using a line (y=ax+b). The parameter of the x-axis is concentration (in pg/ml), the parameter of the y-axis is given as MFI. The MFI of each standard point is blankcorrected by subtraction of the Blank-MFI (MFI-MFI_Blank). This type of analysis allowed to detect a value between the concentration of the lowest standard and corrected Blank-MFI.

The calculation of LLOQ and ULOQ was based on the maximum acceptable Bias with Cut-off at 30% Bias and Fit 100%. So practically we were able to detect values between 0 and lowest standard value (посочени са от производителя) . This change can be found – page 5, paragraph 7, and linе 116-127 in the revised manuscript.

Comments 5:  Statistical values obtained should be included in the text. 

Response 5: Thank you for pointing this out. This information will be added to the text. This change can be found – page 8, paragraph 1, linе 214, 215;  page 9, paragraph 1, line 230; page10, paragraph 2, line 247; page 11, paragraph 1, line 262; page 12, paragraph 1, line 271; page 13, paragraph 1, line 277-280 in the revised manuscript. 

Comments 6:  As mentioned before, besides the authors' included limitations ("The small number of participants and the lack of healthy controls and patients who do not take HART"), it is necessary to study PLWH with matched-history of therapy but without viral suppression.   

Response 6: As far as I can tell, Comments 6 matches the remark from Comments 1. I apologize in advance if I can't tell the difference. It is explained above why patients with matched-history of therapy but without viral suppression were not included in the study. In the text, it will be added that the study does not include PLWH with matched-history of therapy but without viral suppression, in order to make it as clear as possible to the reader what the value of the presented work is.  Тhis change can be found – page 15, paragraph 2, and linе 382, 383 in the revised manuscript.

Minor comments:  а. Check the cART expressions (also appear as "cAPT", "CART"). b. Tables 3, 4, and 5 may be shorten, just including in the text the relevant values and its statistics.

Response :  а. Thank you for pointing this out. We will fix "cAPT" and  "CART" expressions. Тhis change can be found – page 13, paragraph 5, linе 289; page 13, paragraph 9, line 319; page 14, paragraph 4, line 347, page 15, paragraph 2, page 383 in the revised manuscript. b. Thank you for pointing this out. Tables 3, 4, and 5 will be shorten to make the text less cumbersome.

4. Response to Comments on the Quality of English Language

Point 1: Only minor amendments are needed.

Response 1: Sorry for the Quality of English Language errors despite our best efforts. If possible, let us know and we will correct them.

Round 2

Reviewer 1 Report

Comments and Suggestions for Authors

The revised version addressed my questions.